# Good Practices on Endoscope Reprocessing in Italy: Findings of a Nationwide Survey

**DOI:** 10.3390/ijerph191912082

**Published:** 2022-09-24

**Authors:** Michela Scarpaci, Tommaso Cosci, Benedetta Tuvo, Alessandra Guarini, Teresa Iannone, Angelo Zullo, Beatrice Casini

**Affiliations:** 1Department of Translational Research, N.T.M.C. University of Pisa, Via S. Zeno 35-37, 56127 Pisa, Italy; 2Gastroenterology and Digestive Endoscopy, ‘Nuovo Regina Margherita’ Hospital, 00153 Rome, Italy; 3Gastroenterology Unit, ‘Polistena’ Hospital, 89024 Reggio Calabria, Italy

**Keywords:** endoscopes reprocessing, quality assessment, training on reprocessing

## Abstract

**Background**: Correct reprocessing and microbiological surveillance on endoscopes are fundamental for preventing the transmission of multi-drug resistant strains and device-related infections. **Methods**: A questionnaire with three domains was created: (1) centre characteristics; (2) endoscope reprocessing procedures; and (3) application of microbiological surveillance. Nurses working in endoscopic units across Italy were invited to anonymously fill out the questionnaire on the SurveyMonkey platform between November 2021 and February 2022. **Results**: A total of 82 out of 132 endoscopic centres participated in the survey, with at least one centre from each Italian region. Data found different concerns regarding the current practice of both reprocessing and microbiological surveillance. According to respondents, the training on reprocessing was performed through theoretical training and only in 10% of centres; the microbiological surveillance was regularly performed in 59% of centres; and sampled endoscopes were not excluded for use in 31% of centres performing the surveillance until the outcome was pending, and when positive, 72% maintained them in quarantine until a successive negative result. **Conclusions**: Reprocessing and microbiological surveillance currently present several criticisms along the endoscopic centres in Italy. Our survey highlights the need for the correct application of the national recommendations in each endoscopic centre to prevent the potential transmission of endoscope-related infections.

## 1. Introduction

Since the first reported transmission of *Klebsiella pneumoniae* carbapenemase-producing *Enterobacteriaceae* infection (KPC-EI) related to the use of duodenoscopes 10 years ago [1], other cases have been described in literature [2,3,4]. Although the incidence remains cumulatively low, the consequences are clinically relevant and potentially life-threatening for patients [5]. In detail, duodenoscopes were the more frequent culprit for transmitting infections because of their complex structural conformation preventing correct reprocessing and rendering the instrument prone to biofilm formation [6] Indeed, the use of these endoscopes was associated with a higher number of cluster infections as compared to all the other medical or surgical devices [5]. These observations prompted different international societies to produce specific documents with recommendations on procedures for correct microbiological surveillance on endoscopes [6,7,8,9,10,11]. The microbiological surveillance of flexible endoscopes after reprocessing, during storage, or before use is recommended in US standards for duodenoscopes [6]. In Europe, the explicit recommendation to perform microbiological investigation on all endoscopes, not only duodenoscopes, is supported by major scientific societies, such as the European Society of Gastrointestinal Endoscopy (ESGE) and the European Society of Gastroenterology and Endoscopy Nurses and Associates (ESGENA) [7]. According to the practices reported by the cited recommendations, in 2021, an Italian Consensus document on microbiological surveillance after the reprocessing of flexible endoscopes was delivered by different societies of operators involved in the use of endoscopes, in the reprocessing process, and the prevention of healthcare-associated infections [12]. In particular, this document defines the non-conformities in the reprocessing process based on the results of the microbiological surveillance and indicates the corrective actions to be implemented to improve the quality of the process. The procedures suggested in this position paper may be used as a reference standard for comparison to what is performed in routine practice. We, therefore, designed a survey on this topic to assess the current practice in different endoscopic units distributed throughout the entire country. The information from this investigation is useful to identify those aspects requiring implementation. 

## 2. Methods

The survey was commissioned by SIMPIOS (Italian Multidisciplinary Society for the Prevention of Infections in Health Care Organizations) and ANOTE-ANIGEA (National Association of Operators of Endoscopic Techniques-National Association of Gastroenterology Nurses and Associates). In detail, a working group was created with members of both societies who prepared a specific questionnaire with 3 domains focusing on: (1) characteristics of the centre, devices used, and personal training; (2) the procedures followed for endoscopes reprocessing; and (3) the application of microbiological surveillance. 

The questionnaire was created and emended among a selected number of nurses and medical staff of the Endoscopy Units enrolled (15 and 10, respectively) who are experts in reprocessing and endoscopy. The feedback from this group was used to establish content validity and to revise the content of the survey instrument. The wording of the questions was re-examined to see if they contained anything potentially problematic for the reader (double-barrelled questions, negative words, and overly long questions). The reported completion time ranged from 10 to 15 min. 

An invitation letter containing the survey protocol was sent by e-mail to all the Regional Representatives of ANOTE-ANIGEA society, who forwarded it to the endoscopic centres of their region, for a total of 131 recruitments. Volunteer participation was possible through a link to the SurveyMonkey platform accessible from November 2021 to February 2022. Duplicated submissions from the same respondent were prevented. Data were anonymously collected and analysed (Figure 1). The percentages were calculated according to the number of replies available for each question. Answering only a quarter of the questions was considered an exclusion criterion.

A formal ethical approval for this survey study was not required.

## 3. Results

### 3.1. Characteristics of Centres 

Of the 131 recruited centres, a total of 82 centres participated in the survey. For each centre, one member of the endoscopy staff responded on behalf of its endoscopic unit. The respondent centres were located in all regions of Italy except three, with a median of 3 (range: 1–15) centres per region (Figure 1). 

Of all Endoscopy Unit, 50 were located in general hospitals (61% overall), 19 in university hospitals (23%), and the remaining 13 in research institutes (16%). According to respondents, the estimated number of endoscopies performed yearly was 5000–10,000 in the majority (47%) of centres, followed by 2500–5000 (23%), and >10,000 (17%), with less than 2500 in the remaining centres. (Figure 2)

### 3.2. Endoscopes and the Devices Utilized

Information on the type of endoscopes (gastroscope, colonoscope, etc.) and other devices used in the endoscopic unit were available for 61 centres. The respondents reported that endoscope procedures with duodenoscopes were performed in 89% of centres, and with echoendoscopes, enteroscopes, and cholangioscopes in 57%, 38%, and 25%, respectively. The bronchoscopes were used in 18 (29%) centres. Among the 54 centres equipped with duodenoscopes: in 16 (30%), only instruments with a disposable distal cap were used; in 6 (11%), only those with reusable caps were utilised; and in 13 (24%), only instruments with a non-removable cap were used. While in the remaining 19 endoscopy units all three duodenoscopes types were employed. Based on the answer given by respondents, single-use valves for endoscopes were routinely utilized in 11%, whilst their use was tailored (transplant, oncological, and immunocompromised patients or in endoscopies performed in those with a known infection with MDR strains) in 46% of hospitals. Regarding the irrigation bottle used to flush air/water channels during the examination, it was estimated that a reusable bottle was utilised in 68% of centres, and it was daily sterilized in 71% of cases. 

### 3.3. Training on Reprocessing

Regarding the question on specific training for reprocessing (80 replies), respondents stated that only a practical training with the supervision of expert colleagues or the coordinator was performed in 81% of centres. A specific theoretical training before practice was accomplished in 10% of hospitals, whilst 9% of respondents declared that no theoretical training nor formal supervision was completed by nurses performing reprocessing (Figure 3). Regarding professional update on the endoscope reprocessing, a formal retraining was estimated to be performed yearly in only 38% of centres.

### 3.4. Reprocessing Procedures

According to the 60 centres responding to these questions, the reprocessing of endoscopes was performed by dedicated staff in 40 (67%), nurses (12%) in endoscopy units, support personnel (auxiliary nurses) in 68%, and alternatively both figures in the other 20%. The other 20 centres reported that dedicated reprocessing staff was lacking, but operators alternate according to the shift and the department’s needs (Figure 4). While the precleaning procedure was estimated to be performed in 99% of units, the leak test was executed in only 58% of centres. For channel brushing, a single-use device was documented by respondents in 85% of structures. For high-level disinfection, the use of automated washer-disinfector machines was reported by all, but 1 centre where manual disinfection was still performed. Appropriate storage cabinets for reprocessed endoscopes were available in 60% of hospitals. Closed containers were never used. 

### 3.5. Microbiological Surveillance

Information on microbiological surveillance was available from a total of 68 centres, and according to respondents it was regularly performed in 40 (59%) of them. In detail, for duodenoscopes and linear echoendoscopes, the microbiological sampling is reported to be performed monthly, every 3–6 months, or yearly in 32%, 56%, and 12%, respectively, whilst in none was it scheduled based on the number of examinations performed with these instruments. For other endoscopes, the surveillance was indicated to be performed every 1–3 month (18%), every 3–6 months (33%), or yearly (49%). Only 7 (17.5%) centres declared to perform an additional microbiological sampling when the instrument was used in a patient infected with a multi-drug resistance (MDR) strain. While the 52% of respondents reported that the sampling was performed following at least 6 hours from reprocessing, the remaining performed it before this timeframe. The 97% of endoscopy units state that the sampling included the endoscope channels, of which 54% also tested the external surfaces, whilst in the remaining 3% exclusively the water inside of the washer-disinfector machines was tested. The ‘flush-brush-flush’ or the ‘flush-such-flush’ sampling was referred to be adopted by 70% and 15% of centres, respectively, whilst only the ‘flush’ procedure was documented in the remaining 15% units. Sterile water, saline water, or Tween-80 was declared to be used as eluent by 44%, 41%, and 15% centres. The water at entry and exit of the washer-disinfector machines was sampled in 76% hospitals, the internal surfaces of the storage cabinets in 32%, and that of the air-water bottle in 21% of cases. Respondents reported that the sampling procedure involved endoscopy nurses in 62% of centres, personnel of hospital hygiene service in 23% of centres, and by other professional figures in the remaining cases. 

According to the answers, the sampling analyses were performed by the internal microbiological unit in 70%, by internal hygiene unit in 10%, and by external accredited laboratories in 20% of hospitals. 

Following the microbiological sampling, 31% of respondents stated that the instrument was not excluded by use. When the testing was positive, 72% of centres affirmed that the endoscope was maintained in quarantine until the successive negative result. Ten (25%) endoscopy units reported that a revision of the entire procedure was performed, and in eight (20%) centres a retrospective surveillance in patients who underwent endoscopy with the contaminated endoscope was completed. 

## 4. Discussion

This survey mirrors the current practice of both reprocessing and microbiological surveillance on endoscopes in several Italian centres distributed through the entire country. Some relevant concerns emerging from data analysis merit an interpretation in order to improve the quality of procedures and prevent the spread of infection with endoscopes. The reprocessing of endoscopes is a complex process, and the correct execution of all phases is crucial to minimize the risk of residual contamination. This particularly applies to duodenoscopes and echo-endoscopes whose structural conformation is challenging for cleaning and disinfection. Therefore, the endoscopy staff need specific training to be able to perform the reprocessing in a correct and accurate manner on every type of endoscope, according to the manufacturer’s instructions for use and in the position paper. 

Adequate formal training is, therefore, crucial to ensure the quality of reprocessing procedures [13]. Our data found criticism in this field, because only 10% of the respondents affirm to provide specific theoretical and practical training for endoscopy staff. Moreover, 9% of centres reported that the staff performing reprocessing was not supposed to be trained through specific theoretical training nor a practical period with direct supervision by an expert until the skill is achieved. Overall, the data found that the execution of the pre-cleaning and high-disinfection phases of reprocessing agreed with recommendations in the position paper. However, we observed that, according to respondents, the leak test before reprocessing was not performed in 28 (39%) centres, despite it being a critical procedure to avoid instrument damage during immersion [14].

Regarding the use of appropriate devices, the data found that less than half (48%) of the participating centres currently use duodenoscopes with removable caps. A removable cap allows for a better manual cleaning and brushing of channels and its use is strongly advised [15,16].

As far as microbiological surveillance is concerned, the data found that it was performed in only 59% of endoscopic units. Moreover, the sampling of duodenoscopes and echoendoscopes was performed only every 3–6 months in more than half of the centres, rather than monthly or earlier when more than 60 procedures were performed with the same instrument, as recommended [8,12]. It was relevant to note that only 21% of centres reported performing an additional microbiological sampling when the endoscope was used in patients with known MDR strain infection. This is particularly critical considering the risk of potential cross-infection and the diffusion of resistance genes among bacteria within the biofilm of contaminated endoscopes [17]. Moreover, 45% of respondents affirm that the sampling was performed early than 6 hours following reprocessing and with sterile water—rather than saline or Tween-80—as suggested by guidelines, reducing the microbial recovery rate [18,19].

Another relevant finding emerging from this investigation was that the endoscopes subjected to the microbiological investigation were not quarantined until the outcome of the culture in 12.1% (4/33) of centres, different from what was suggested in the guidelines [8,12].

Despite the several criticisms found, this survey allowed for attention to be placed on the adoption of inappropriate practices in reprocessing and microbiological surveillance that require the implementation of corrective actions. The standardisation of the reprocessing practices is particularly challenging in Italy, where each region is regulated by an autonomous health system. 

This was the first national survey conducted in Italy among a representative number of endoscopy units. 

## 5. Conclusions

In conclusion, the data of this first survey involving endoscopic units distributed across Italy showed some relevant criticisms on the execution of the reprocessing procedures and microbiological surveillance. In detail, some concerns remain in several fields among the participating endoscopic units, such as the training, inappropriate procedures of reprocessing, and lack of uniformity of criteria for definition and corrective measures in the case of non-compliant endoscopes. The effort made in delivering the Italian Consensus Document was aimed to standardize the practice of microbiological surveillance on endoscopes and reduce the heterogenicity of practices among all Italian Regions, thus, to improve the reprocessing quality and safety of patients. The proposed questionnaire was not focused on the results of microbiological surveillance among the responding endoscopic units. However, for the next survey, which will be designed to evaluate the implementations made by the same endoscopy units, it would be interesting to include this topic, especially in terms of the contamination rates of instruments and the type of MDROs, with the comparison of results between centres. Our observational survey highlights the need for the correct application of the national recommendations in each endoscopic centre to prevent the potential transmission of endoscope-related infections. 

## Figures and Tables

**Figure 1 ijerph-19-12082-f001:**
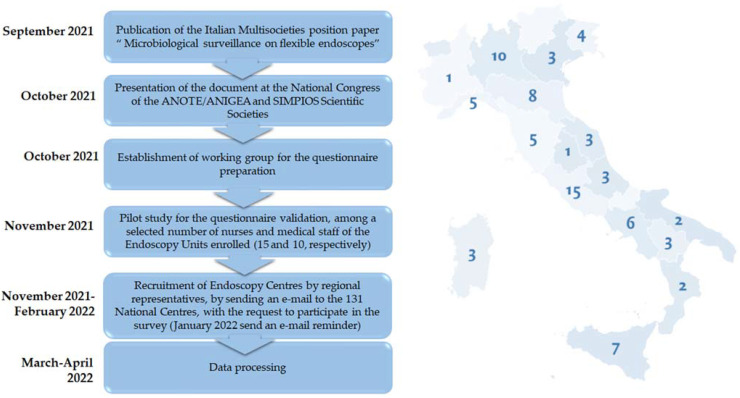
Timeline of the study design and geographical distribution of respondent centers per region.

**Figure 2 ijerph-19-12082-f002:**
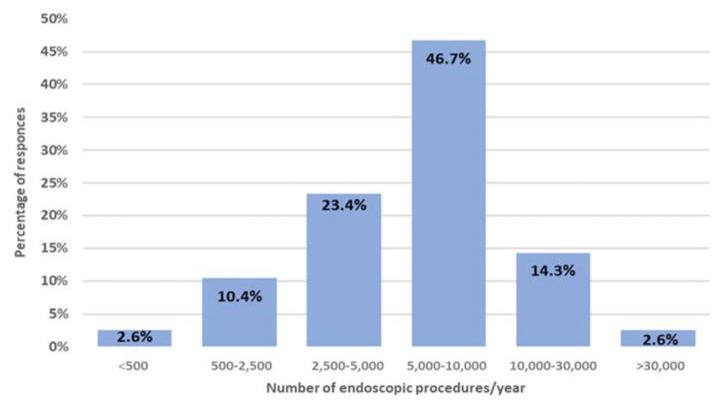
Percentage distribution of the annual endoscopic procedures performed in all 82 respondent endoscopic centers.

**Figure 3 ijerph-19-12082-f003:**
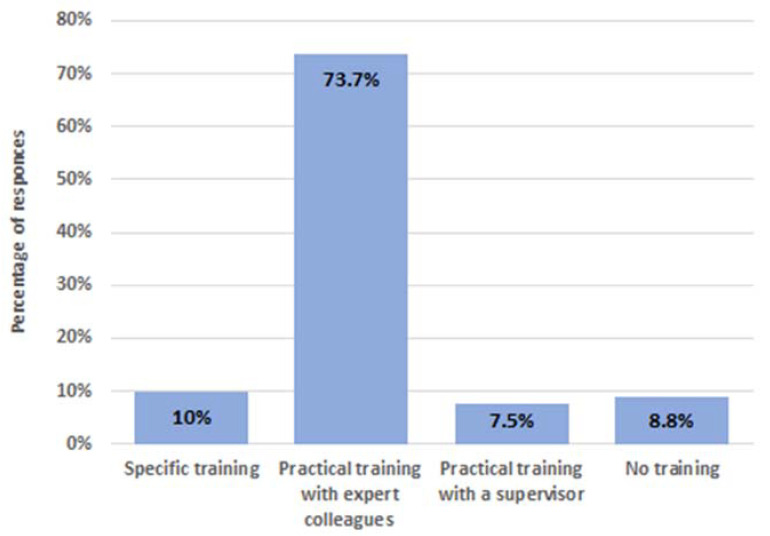
Percentage distribution of the training methods carried out in the respondent endoscopic centers (80 responders).

**Figure 4 ijerph-19-12082-f004:**
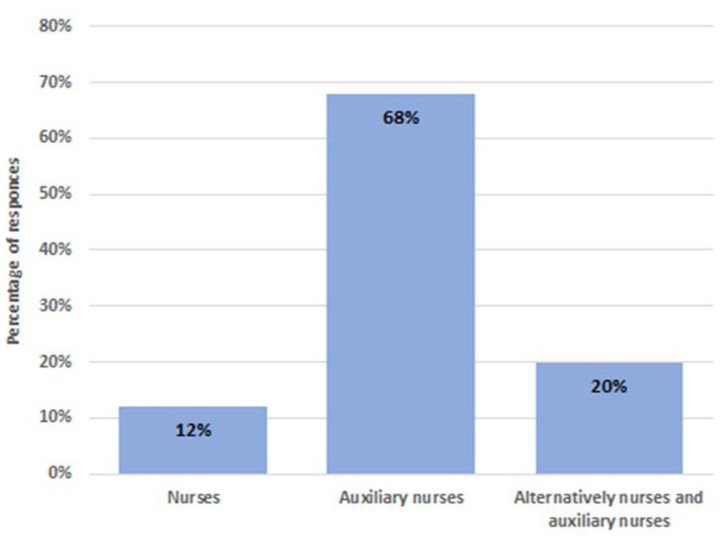
Percentage distribution of all professionals dedicated to reprocessing (40 respondents).

## Data Availability

The data were collected and processed directly by the authors, so we have no link to suggest.

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
