# Peer review of "Good Practices on Endoscope Reprocessing in Italy: Findings of a Nationwide Survey"

_ijerph, 2022, doi:10.3390/ijerph191912082_

Round 1

Reviewer 1 Report (Previous Reviewer 2)

Dear Authors

Greetings

Congrats for all efforts. 

However, I believe you can improve your work by following the suggestions I sent earlier, including graphics, conclusion and references.

Regards

Author Response

Thanks for your review. We have taken note of your comments, which we share to a large extent. We have improved the figures and the conclusions section as suggested
All changes are highlighted in red in the new version of the manuscript and the figures have been replaced.

Best regards

Reviewer 2 Report (Previous Reviewer 3)

The authors corrected all the issues. The corrections are shown transparent in the manuscript. There are no further issues. Therefore, I recommend publication.

Author Response

Thanks for your review

Best regards

Reviewer 3 Report (New Reviewer)

nice study on the current practice of scope sterilization, which shows that there is much room for improvement. This is an important message, and should be a wake-up call for endoscopic centers

Author Response

Thanks for your review

Best regards

This manuscript is a resubmission of an earlier submission. The following is a list of the peer review reports and author responses from that submission.

Round 1

Reviewer 1 Report

Dear Authors

Some corrections have been made; however, I think the graphs and presentation still need to improve. Also, a flow chart or schematic illustration of the study procedures could make your article more appropriate. 

Reviewer 2 Report

Dear Authors,

Greetings

Thank you for all efforts to improve your article.

Kind regards

Reviewer 3 Report

The manuscript Endoscope reprocessing and microbiological monitoring practices in Italy: findings of a nationwide survey is an interesting paper with some scientific importance. The manuscript has potential for publication consideration but before that, some issues must be solved:

Title: Correct the title like an Opinion poll on endoscope reprocessing… Since you did not analyse practices or the microbiological analysis. The title is very promising, but the results are simple surveys.

Introduction: the text is scarce. It looks like an abstract. The authors should add the text on standard procedures of endoscope disinfection. What is “the gold standard”, especially as you are comparing results in discussion to the Italian consensus document? Also is this Italian document similar or different from other EU countries guidelines? If yes, why? If not, you can make a comparison to other EU countries. The authors should upgrade the introduction part. 

Methods: Did you include all 20 Italian regions? If so, It would be great to show the distribution of respondents per region.  

Terminology: the authors must be cautious when interpreting the results. All the interpretations must be in the direction like respondents reported, respondents believe, respondents think, … etc. For example, the number of endoscopies performed is not exact data, but an estimation of the respondents. And so must be the interpretation of the results. The same about the training methods. Line 181: it is wrong to say the precleaning procedure was performed in 99%, but rather the respondents reported that the precleaning procedure was performed in 99%. The entire abstract, results, discussion and especially conclusions must be rewritten in the correct way.  

Results: upgrade the manuscript by adding demographic data of the respondents, e.g., age, gender, occupation, region… Provide some comparison. Simple answer frequencies are not enough for scientific publications.  

Line 78: The authors must specify what is the representative sample for each Italian region and how they set the threshold.

Figure 1: The authors must correct the y-axis. There is no need for two decimals in the figure. Also, change the decimal comma to the decimal point. Add the size of the sample (n) to the figure.

Figure 2: see comments on figure 1

Figure 3: see comments on figure 1. The sum of all answers is 116.4% The authors must comment on this.

Section 3.5: results of the microbiological analysis would increase the interest in this paper. Right now, all we know is that hospitals maintain the endoscope and they test the quality of disinfection. What are the results? Which strains were found on the surface of endoscopes? What was the procedure for total eradication? This entire part is missing in section 3.5.

Line 196: correct the decimal comma to point.

General: Did you provide any kind of statistic (even the simplest one)? You stated in conclusions that you don’t have results of microbiological analysis, nor do I see any kind of comparison, analysis, or something similar. Just to show frequencies of survey answers is low-quality paper. I recommend whether you upgrade your paper in comparison survey with the result of microbiological analysis of endoscope or provide intensive statistical analysis of your data.